METHODS AND RESOURCES

# Precise genomic mapping of 5-hydroxymethylcytosine via covalent tether-directed sequencing

**Povilas Gibas** [ID][☯], **Milda Narmontė** [☯], **Zdislav Staševskij, Juozas Gordevičius, Saulius Klimašauskas** *, **Edita Kriukienė** [ID]*

Institute of Biotechnology, Life Sciences Center, Vilnius University, Vilnius, Lithuania

☯ These authors contributed equally to this work.
* saulius.klimasauskas@bti.vu.lt (SK); edita.kriukiene@bti.vu.lt (EK)

**Data Availability Statement:** Raw and processed hmTOP-seq data generated in this study have been deposited in the NCBI Gene Expression Omnibus under accession number GSE140206.

## Abstract

5-hydroxymethylcytosine (5hmC) is the most prevalent intermediate on the oxidative DNA demethylation pathway and is implicated in regulation of embryogenesis, neurological processes, and cancerogenesis. Profiling of this relatively scarce genomic modification in clinical samples requires cost-effective high-resolution techniques that avoid harsh chemical treatment. Here, we present a bisulfite-free approach for 5hmC profiling at single-nucleotide resolution, named hmTOP-seq (5hmC-specific tethered oligonucleotide–primed sequencing), which is based on direct sequence readout primed at covalently labeled 5hmC sites from an in situ tethered DNA oligonucleotide. Examination of distinct conjugation chemistries suggested a structural model for the tether-directed nonhomologous polymerase priming enabling theoretical evaluation of suitable tethers at the design stage. The hmTOP-seq procedure was optimized and validated on a small model genome and mouse embryonic stem cells, which allowed construction of single-nucleotide 5hmC maps reflecting subtle differences in strand-specific CG hydroxymethylation. Collectively, hmTOP-seq provides a new valuable tool for cost-effective and precise identification of 5hmC in characterizing its biological role and epigenetic changes associated with human disease.

## Introduction

DNA methylation is involved in many biological processes such as embryogenesis, establishment of cell identity and organismal fate, and development of various pathological conditions, including cancer. A well-documented repressive role of 5-methylcytosine (5mC) can be reversed via the action of the Ten-Eleven Translocation enzymes (TET1, 2, and 3), which remove 5mC through the formation of several oxidized forms of 5mC: 5-hydroxymethylcytosine (5hmC), 5-formylcytosine, and 5-carboxylcytosine. In addition to being a 5mC demethylation intermediate, a number of studies have reported 5hmC as a stable epigenetic mark with biological role in transcription regulation in both physiological and pathological states [1–6]. Of all known oxidized forms of 5mC, 5hmC is the most abundant in mammalian tissues,

**Funding:** The work was supported by the Research Council of Lithuania (https://www.lmt.lt/en/) (Researcher groups project No. MIP-58-17 to EK) and the European Research Council (https://erc.europa.eu/) (ERC-AdG-2016/742654 to SK). The funders had no role in study design, data collection and analysis, decision to publish, or preparation of the manuscript.

**Competing interests:** I have read the journal's policy and the authors of this manuscript have the following competing interests: SK, ZS, and EK are inventors on a patent related to the TOP-seq profiling strategy of genomic sites.

**Abbreviations:** 5hmC, 5-hydroxymethylcytosine; 5hmCG, hydroxymethylated CG site; 5hmCH, hydroxymethylated CH site; 5mC, 5-methylcytosine; ABA-seq, DNA modification–dependent AbaSI restriction coupled with sequencing; BGT, β-glucosyltransferase; CGI, CG island; CNV, copy number variation; DBCO, dibenzocyclooctyne; eM.SssI MTase, an engineered version of the CG-specific DNA cytosine-5 methyltransferase M.SssI; gDNA, genomic DNA; H3K27ac, histone H3 lysine 27 acetylation; H3K36me3, histone H3 lysine 36 trimethylation; H3K4me1, histone H3 lysine 4 monomethylation; H3K4me3, histone H3 lysine 4 trimethylation; H3K9ac, H3 lysine 9 acetylation; hMe-Seal, 5hmC selective chemical labeling; hmTOP-seq, 5hmC-specific tethered oligonucleotide–primed sequencing; KOD polymerase, the thermophilic DNA polymerase from *Thermococcus kodakaraensis*; LTR, long terminal repeat; mCG, methylated CG site; mESC, mouse embryonic stem cell; M.HhaI MTase, the GCGC-specific DNA cytosine-5 methyltransferase HhaI; NHS, N-hydroxysuccinimide; nt, nucleotide; ODN, oligodeoxyribonucleotide; OR, odds ratio; PDB, Protein Data Bank; Pfu polymerase, the thermophilic DNA polymerase from *Pyrococcus furiosus*; Pvu-Seal-seq, DNA modification–dependent 5hmC enrichment coupled with sequencing; SINE, short interspersed nuclear element; TAB-seq, Tet-assisted bisulfite sequencing; TET, Ten-Eleven Translocation enzyme; TOP-seq, tethered oligonucleotide–primed sequencing; TTS, transcription termination site; uCG, unmodified CG site; uTOP-seq, uCG-specific tethered oligonucleotide–primed sequencing; UTR, untranslated region; WGBS, whole-genome bisulfite sequencing.

whose levels vary in a tissue-specific manner, reaching up to 1.8% of total cytosine in human neurons [7–11]. In various solid tumors, 5hmC was found at significantly reduced levels, reinforcing the diagnostic and prognostic significance of 5hmC in cancer research [12–14]. However, as whole-genome sequencing and data analysis still remain hardly accessible to large-scale population and clinical studies, cost-effective sensitive techniques are in high demand for unlocking the epigenetic role and diagnostic potential of DNA hydroxymethylation marks.

Enzymatic, chemical treatment–based, or antibody enrichment–based technologies have been developed for analysis of 5hmC genome-wide [2,15–19]. Single-nucleotide-resolution whole-genome bisulfite sequencing (WGBS) and its derivatives offer unprecedented capability to infer absolute DNA modification levels [20–23]. However, noticeable degradation of DNA and high sequencing depths required for confident determination of scarcely abundant modifications [24] demand sizeable amounts of input DNA and significant experimental/computational resources, which limit widespread application of WGBS in large-scale clinical and populational studies.

Methods based on covalent 5hmC modification have revealed their potential in genome-wide analysis of 5hmC [2]. However, despite the high sensitivity arising from covalent 5hmC derivatization, all methods employing affinity enrichment, including hMe-Seal [2], suffer from low resolution (200–500 bp). Acknowledging the constraints of the existing methods and inspired by the success of our recently developed uTOP-seq (uCG-specific tethered oligonucleotide–primed sequencing) strategy for single-base-resolution mapping of unmodified CG sites (uCGs) [25], we went on to develop a high-resolution bisulfite-free method for analysis of 5hmC, named hmTOP-seq (5hmC-specific tethered oligonucleotide–primed sequencing). We examined a series of chemo-enzymatic tethering strategies for their suitability to support the tethered oligonucleotide–primed sequencing (TOP-seq) reaction and suggested a first predictive model for this unconventional mode of polymerase action: nonhomologous proximity-driven internal priming. Based on this knowledge, we optimized and validated the developed procedure on model DNA systems and mouse embryonic stem cells (mESCs).

## Results

### Covalent derivatization and sequence readout at 5hmC sites

The hmTOP-seq analysis relies on a previously uncharacterized feature of DNA polymerases —proximity-driven nonhomologous priming to produce adjoining DNA molecules for downstream sequencing and genomic mapping. This event is brought about by a DNA oligonucleotide probe covalently tethered at target sites in DNA. As originally described for TOP-seq analysis of unmodified CGs [25], the covalent tethering of 5hmC residues was achieved in two steps: chemo-enzymatic derivatization of 5-hydroxyl-group with a reactive chemical group followed by selective conjugation of an appropriately derivatized oligodeoxyribonucleotide (ODN). We explored the effects of structural features of the chemical tether on the efficiency of the reaction by Pfu polymerase. In particular, we compared three tethering strategies in terms of their capacity to efficiently and accurately prime the polymerase reaction on DNA (Fig 1A). The first two exploited the bacteriophage T4 β-glucosyltransferase (BGT)-directed transfer of azide-containing chemical moieties to 5hmC from a modified cofactor, UDP-6-azidoglucose [2], but differed in that a copper(I)-catalyzed (Fig 1A, compounds **1a,b**) or copper-free azide-alkyne cycloaddition (compounds **2a,b**) coupling of the ODN was used in the second step. The third strategy (compound **3**) was based on the capacity of the eM.SssI MTase (an engineered version of the CG-specific DNA cytosine-5 methyltransferase M.SssI) to covalently modify 5-hydroxymethylated CG sites (5hmCGs) with aliphatic thiols such as cysteamine (2-mercaptoethylamine), permitting subsequent tethering of an N-hydroxysuccinimide

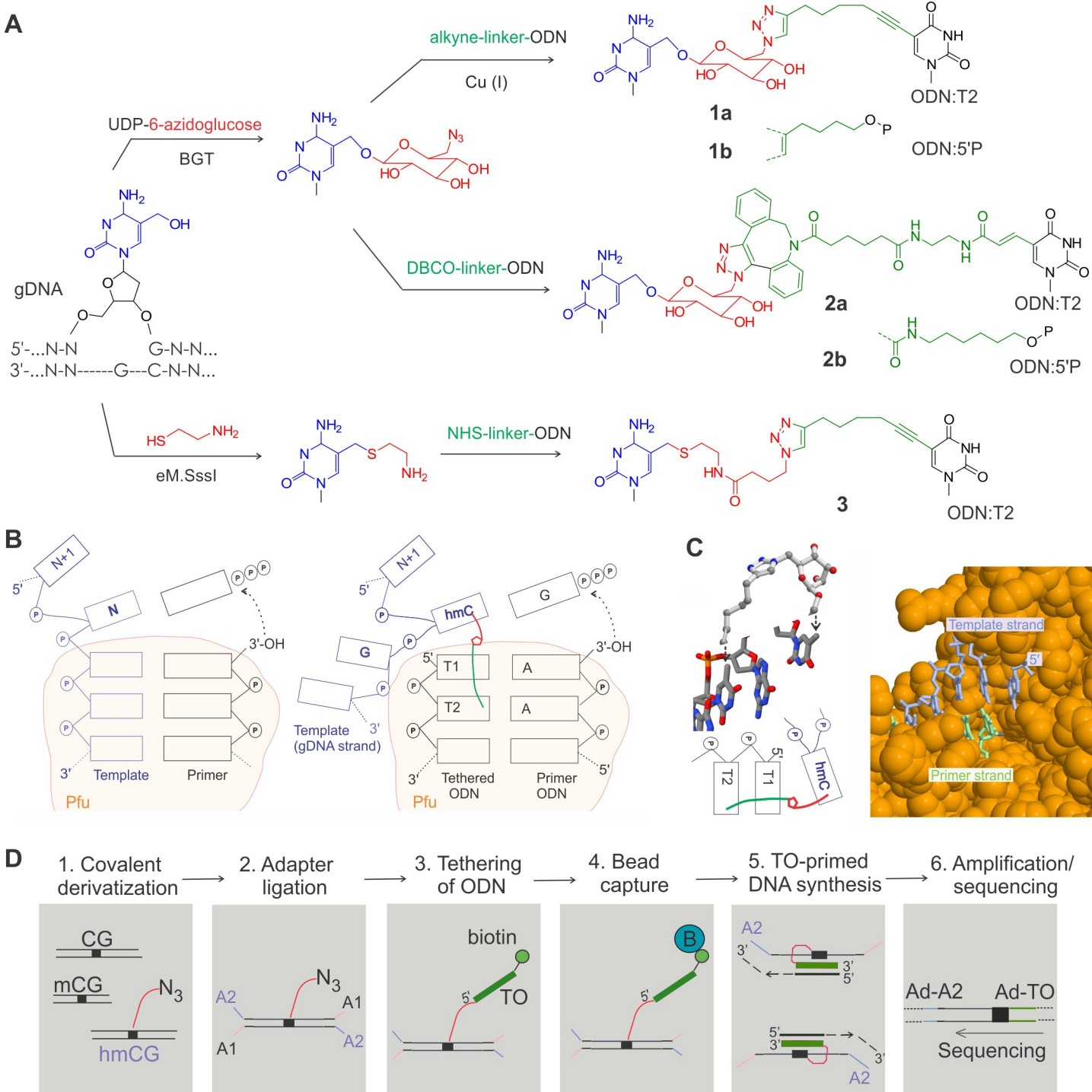

**Fig 1. Tethered oligonucleotide priming and analysis of 5hmC.** (A) Strategies of 5hmC derivatization. (B) Schematic of a conventional primer extension reaction by a DNA polymerase derived from Pfu-DNA cocrystal structure (PDB: 5OMF) (left) and a putative mechanism of tethered ODN–primed polymerase reaction (right). The chemical tether (shown as green/red line) connecting T2 in the bound DNA duplex and the 5hmC residue in gDNA strand facilitates the capture of the 5hmC in a stacked position required for the priming reaction. Beige areas show regions of extensive contacts between the Pfu polymerase and the bound DNA. (C) Left, chemical structure of the tethering linker **1** (top) derived by MM2 conformational energy minimization (S1 Fig) is shown in the context of the template DNA strand 5′..T$_4$G$_5$T$_6$.. of the KOD polymerase–DNA complex (PDB: 4AIL) resembling the positions of the hmC, T1, and T2 nucleotides (bottom) in Fig 1B. Right, a large cleft of Pfu polymerase (space-fill orange) seen from the major groove side of the bound DNA duplex (blue and green strands). (D) Workflow of the hmTOP-seq procedure. (Step 1) Fragmented gDNA is tagged with an azide group through BGT-glucosylation. (Step 2) Azide-modified DNA is ligated to partially complementary adaptors. (Step 3) The ODN containing a

biotin group is tethered to azide groups using click chemistry. (Step 4) The biotin-labeled fragments are captured on streptavidin beads. (Step 5) TO-primed strand extension. (Step 6) PCR amplification with Ad-A2 and Ad-TO primers containing NGS platform-specific 5′-end adaptor sequences. Unidirectional sequencing from the A adaptor sequence included in the 5′ part of Ad-TO-barcode amplification primer. 5hmC, 5-hydroxymethylcytosine; A1 and A2, strands of a partially complementary adaptor; Ad, extended sections of platform-specific adapters; BGT, β-glucosyltransferase; DBCO, dibenzocyclooctyne; eM.SssI, an engineered version of the CG-specific DNA cytosine-5 methyltransferase M.SssI; gDNA, genomic DNA; hmC, 5-hydroxymethylcytosine; hmCG, hydroxymethylated CG site; hmTOP-seq, 5hmC-specific tethered oligonucleotide–primed sequencing; KOD, the KOD DNA polymerase from *Thermococcus kodakaraensis*; mCG, methylated CG site; NGS, next generation sequencing; NHS, N-hydroxysuccinimide; ODN, oligodeoxyribonucleotide; PDB, Protein Data Bank; Pfu, the Pfu DNA polymerase from *Pyrococcus furiosus*; TO, tethered oligodeoxyribonucleotide.

(NHS) ester–containing ODN via the attached terminal amino group [26]. Altogether, this gave us the opportunity to explore chemical linkers of different length, rigidity, and overall bulk. We also examined two different tether attachment points in the ODN strand (S1 Table and S1 Fig). We found that the **1a** and **3** labeling chemistries were well compatible with the nonhomologous priming at or near 5hmC residues in our testing system composed of two model DNA fragments (S2A, S2B and S2C Fig). However, in both cases, higher yields of the desired products were consistently obtained when the tether was linked at the 5-position of the second T residue rather than the 5′-phosphate group (S1 Table). In contrast to the Cu(I) click–derived product, the ODN conjugated through the dibenzocyclooctyne (DBCO) cycloaddition **2a,b** failed to generate priming products of comparable quality and quantity in our hands (S2D Fig). A recent application of the DBCO chemistry (apparently tethered to the 5′-phosphate group as in **2b**) for 5hmC analysis included a lengthy conjugation time (24 h) but ultimately showed a lower accuracy of the priming reaction, which limited the resolution of the method to ±20 nucleotides (nt) [27]. Our developed procedure based on the copper(I) click coupling (compound **1a**) generated sequence reads that predominantly included the full sequence of the tethered ODN immediately followed by the target nucleotide and adjoining sequence (. . .AAAGNNN . . ., see Fig 1B).

Based on these observations and available cocrystal structures of related Pfu and KOD polymerases in complex with their DNA substrates [28,29], we proposed a model for the putative priming complex. The crystallographic evidence indicates that the polymerases tightly bind the duplex part of the DNA, whereas the 5′ part of the template strand is less structured and can assume multiple conformations (S1 Fig). The template-dependent addition of an incoming nucleotide at the 3′ end of the priming strand occurs upon stacking of the template nucleotide with the 5′-terminal nucleotide in the bound duplex (Fig 1B). Similarly, in the case of the tethered-ODN priming, this can be achieved if the ODN duplex binds in the active site of the enzyme bringing the tethered nucleotide of the template DNA strand (5hmC) into spatial proximity and whereby promoting its binding in a stacked position in lieu of the template nucleotide. Although this conformation of the tethered DNA strand may be not the most preferred in general, it has a certain probability to occur (depending on particular structure of the linker) upon prolonged initial amplification cycles, leading to primer strand extension and successive stepwise translocation of the duplex away from the active site. A wide cleft of Pfu transversing the contour of the bound duplex DNA appears to be well suited to accommodate fairly bulky structural extensions on the major groove side of the template strand, such as the downstream part of the template strand and the tethering linkers of a certain size (Fig 1C). Consistent with our experimental observations (S1 Table), MM2 conformational modeling studies suggested though that accommodating the DBCO coupling linker **2a** might be problematic because of its bulkiness and/or steric clashes with the bound DNA duplex (S1 Fig).

## Evaluation of hmTOP-seq on model genome and mESCs

Since the BGT Cu(I)-catalyzed click labeling using **1a** showed best performance as compared with the other tether chemistries (S2 Fig, S1 Table), it was selected for further optimization as part of the ultimate hmTOP-seq procedure (Fig 1D). The sensitivity and specificity of hmTOP-seq was first assessed on a model bacteriophage lambda genome that was pre-hydroxymethylated at the GCGC sites (215 sites in the 42-kb DNA) to various extents using our previously discovered atypical reactivity of the bacterial M.HhaI MTase (the GCGC-specific DNA cytosine-5 methyltransferase M.HhaI) ([26,30] and S2E Fig). All the GCGC sites were detected by hmTOP-seq and correlation between two technical replicates of each library type was very high (Pearson mean r = 0.98; sd = 0.003; $p < 2.2 \times 10^{-16}$) (Fig 2A). Moreover, we observed increasing median read coverage with increasing hydroxymethylation of GCGC sites, which demonstrated quantitative detection of 5hmC by our method (quadratic regression; adjusted $R^2 = 0.9057$; $p = 1 \times 10^{-4}$; Fig 2B). At other CG sites, the median read coverage was a constant zero. Importantly, the majority of mapped reads started precisely at GCGC sites, with a minor fraction distributed in adjacent positions ($-1 \div -4$, S3A Fig), which altogether makes 98% of total reads identifying hydroxymethylated GCGC sites.

We next sought to examine the ability of hmTOP-seq to localize 5hmC residues in mammalian genomic DNA (gDNA). We created base resolution hmTOP-seq maps of mESCs in two replicates using various input DNA amounts (5, 50, and 500 ng) (for sequencing statistics, see S2 Table). We also prepared control hmTOP-seq libraries using the same pipeline without the BGT labeling step. Similarly to the lambda DNA experiment, the majority (93%) of mapped reads started at or in the immediate vicinity of CG sites, with only a fraction of reads distributing around CGs (Fig 2C and S3B Fig). Such a precise readout of CG sites clearly demonstrates the single-base resolution capacity of the method. Replicates of the higher-input hmTOP-seq libraries correlated well (Pearson mean r was 0.46 and 0.8 for 50-ng and 500-ng input libraries, respectively) (Fig 2D), and notably, subsampling of the original datasets down to 30% (approximately 9.6 million and 6.7 million processed reads, respectively) only marginally influenced the correlation (S4 Fig), suggesting that the sequencing depths could be lowered considerably. The technical replicates of 5-ng DNA input libraries showed considerably smaller correlation (Pearson r = 0.11).

With the 500-ng and 50-ng input DNA samples, we found approximately 4.8 million and 2 million 5hmCGs, respectively, that overlapped between replicates of the libraries (mean coverage 5.7× and 7×, respectively). Only about 0.26 million overlapping 5hmCGs were detected in the 5-ng input samples (mean coverage 13.7×). Taken together, these results showed that reducing the input DNA library size leads to an increased variability even at high sequencing depths. However, the variability dropped and correlations increased after calculation of a normalized density (h-density) for 180-bp windows as described for uTOP-seq [25] for all input DNA libraries (Fig 2D). Importantly, the 500-ng control libraries had mean coverage 1.5× and showed very low correlation (Pearson r = 0.03), which indicates that hmTOP-seq detects 5hmCG modification even at very low DNA inputs.

We then compared our 5hmCG datasets with the bisulfite treatment–based TAB-seq data [18] and first calculated the overlap of 5hmCGs detected by TAB-seq (1.9 million CGs) and hmTOP-seq. hmTOP-seq recovered 50% and 25% of TAB-seq–identified 5hmCGs in the 500-ng and 50-ng input DNA datasets, respectively (odds ratio [OR] = 4 and OR = 3.8, respectively, $p < 2.2 \times 10^{-16}$; Fisher's exact test). Notably, the amounts of identified 5hmCGs in 500- and 50-ng input DNA hmTOP-seq libraries outnumbered those obtained by TAB-seq, which normally requires micrograms of starting DNA. Comparison between the hmTOP-seq

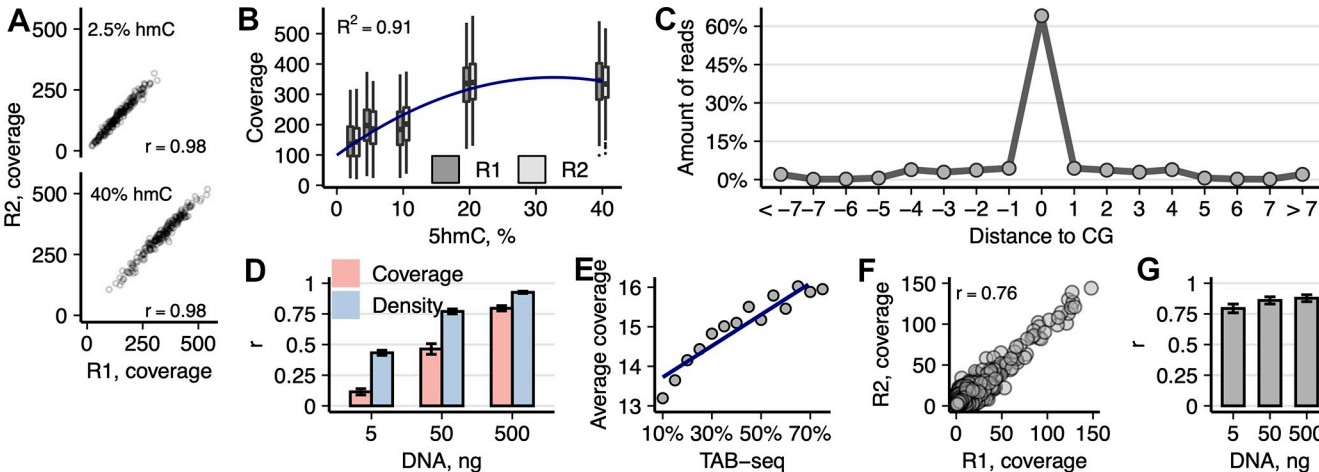

**Fig 2. Analysis of hmTOP-seq libraries in model lambda genome and mESCs.** (A) Correlation between the coverage of 2.5% and 40% pre-hydroxymethylated GCGC sites in the technical replicates of hmTOP-seq libraries of lambda bacteriophage genome. (B) Dependence of hmTOP-seq coverage on the level of hydroxymethylation of GCGC sites in bacteriophage lambda DNA. Quadratic regression was used to fit the plotted data ($Y = 99.5 + 15.7X - 0.240X^2$). (C) Distance distribution of read start positions from a nearest CG site in the hmTOP-seq library of mESC DNA (500-ng input). (D) Correlation of 5hmCG coverage and h-density signals in replicates of hmTOP-seq libraries prepared with varying amounts of mESC DNA. h-density was computed by normalizing coverage values by the unweighted CG density as described in [25]. (E) Comparison of hmTOP-seq coverage and 5hmC percentages estimated by bisulfite-based TAB-seq. Each dot represents the average hmTOP-seq value for specific TAB-seq percentage group (97% of all CG that overlap between hmTOP-seq and TAB-seq are used for analysis). (F) Correlation between hmTOP-seq coverage at 5hmCHs in technical replicates of 500-ng input mESC DNA libraries (OR = 347, $p < 2.2 \times 10^{-16}$; Fisher's exact test). (G) Correlation of 5hmC signal between mESCs hmTOP-seq and nano-hmC-Seal data (average peak region size 615 bp). Within each nano-hmC-Seal peak region, total amount of signal from both methods was square-root transformed and correlated per each autosome. The data underlying this figure are included in S1 Data. 5hmC, 5-hydroxymethylcytosine; 5hmCG, hydroxymethylated CG site; 5hmCH, hydroxymethylated CH site, where H = A, C, or T; hmTOP-seq, 5hmC-specific tethered oligonucleotide–primed sequencing; mESC, mouse embryonic stem cell; OR, odds ratio; TAB-seq, Tet-assisted bisulfite sequencing.

coverage and the percentages of 5hmC estimated by TAB-seq in the overlapping set of CGs indicated a good agreement between the two methods (Fig 2E).

The inherent single-nucleotide resolution of hmTOP-seq and sequence nonselectivity of BGT [2] opened a possibility for 5hmC identification in non-CG context. In mESCs, cytosine methylation and hydroxymethylation is present at CHG and CHH sites (H = A, C, or T) [18,31]. Using six hmTOP-seq control libraries, we observed 55,025 non-CG sites of which only 190 overlapped in at least two control libraries, and only 284 of them overlapped with the hydroxymethylated CH sites (5hmCHs) (76,665 occurrences; see below) that we identified in the 500-ng hmTOP-seq libraries, suggesting that these sites resulted from a random priming events rather than from BGT-directed covalent labeling and could be defined as false positive. Moreover, correlation between the control libraries was close to zero, for example, r = 0.03 for the 500-ng libraries. Of all 5hmCHs detected in the 500-ng hmTOP-seq libraries, for further analysis we selected only those sites which overlapped between technical replicates, resulting in a final set of 76,665 5hmCHs (mean coverage 2.7×; Pearson r = 0.76) (Fig 2F). In comparison, CH positions that did not overlap had lower 1.3× sequencing counts ($p < 2.2 \times 10^{-16}$, two-sided $t$ test). Of note, as sequencing counts indicate relative hydroxymethylation of cytosine, lower 2.7× average coverage of the called 5hmCHs in relation to 5hmCGs (5.7×) points to a lower 5hmC occupancy at 5hmCHs. Fifty percent of detected 5hmCHs were found in CA sites (CA:CT:CC = 0.50:0.33:0.17) and distributed in a ratio 1:2 in CHG and CHH context, respectively (34% and 66%). These proportions corresponded well to those reported by the restriction enzyme–based 5hmC profiling methods ABA-seq and Pvu-Seal-seq [19,32]. The total number of the hmTOP-seq called 5hmCHs constituted 1.6% of 5hmCGs. This is close to 1.3% reported by TAB-seq [18] but differs slightly from the data presented by ABA-seq and

Pvu-Seal-seq (4.1% of 5hmC exist in CH context [1.4% CHG and 2.7% CHH]). Similarly to the above-mentioned restriction enzyme–based methods, we identified more 5hmCHs than TAB-seq (26,616 CH positions were detected by TAB-seq) and generally confirmed the presence of 5hmC at non-CG sites in mESCs. Overall, all these data indicate a good reproducibility and capacity of hmTOP-seq to localize genomic 5hmC sites in gDNA.

Next, we performed a pairwise comparison between hmTOP-seq and nano-hmC-Seal, a low-resolution method that also adds an azide-modified glucose moiety to 5hmC and then pulls down modified DNA fragments without preserving single-CG hydroxymethylation information [33]. We observed a considerable agreement between the two types of data for all hmTOP-seq datasets (Pearson mean r = 0.88; sd = 0.03; $p < 2.2 \times 10^{-16}$), despite different resolutions of the two methods (average used nano-hmC-Seal region sized was 615 bp) (Fig 2G).

The analysis of 5hmCG distribution at various genomic elements demonstrated good agreement with the published data [18,19,34]. The highly hydroxymethylated CG sites (top 20% of hmTOP-seq data) were enriched in poised enhancers marked by histone H3 lysine 4 mono-methylation (H3K4me1) histone marks, active enhancers (marked by histone H3 lysine 27 acetylation [H3K27ac] and histone H3 lysine 4 trimethylation [H3K4me3]), exons, 3′ untranslated regions (UTRs), downstream regions of protein-coding genes, shores of CG islands (CGIs), and nonactive promoters depleted in histone H3 lysine 9 acetylation (H3K9ac) histone mark (Fig 3A). CGIs, active promoters marked by H3K9ac, intergenic regions, and all major type of repeats were depleted in 5hmCGs (data shown only for long terminal repeats [LTRs]), except for short interspersed nuclear elements (SINEs) that demonstrated moderate enrichment for less hydroxymethylated 5hmCGs. Along a composite gene, 5hmCGs were most abundant in introns and their density increases toward the 3′ end of the gene (Fig 3B). Because of a small number of identified 5hmCHs, we were unable to accurately estimate the metagene profile of 5hmCHs, even though 53% of identified sites resided in protein-coding genes (S5 Fig) and showed enrichment towards the sense strand (OR = 1.1, $p = 5.3 \times 10^{-8}$; Fisher's exact test).

## Strand-specific analysis of 5hmCGs at gene bodies

As hmTOP-seq targets both individual strands of a CG dinucleotide separately, it has a capacity to display strand-specific hydroxymethylation information. Following the previous observations that 5hmCGs are asymmetrically modified on different strands [18,35], we analyzed strand-specific distribution of 5hmCGs across gene-associated elements. In TAB-seq analysis, the average abundance of 5hmC at the called CG sites is 20% compared with 10.9% at the opposite cytosine [18]. We detected a shift in CG hydroxymethylation toward the sense strand relative to the antisense strand in gene bodies. Importantly, the extent of the strand-specific 5hmCG bias increased according to the expression levels of genes, showing the strongest bias for the highly expressed gene group ($p = 0.4$ and $p = 1 \times 10^{-63}$, for low- and high-expression groups, respectively, two-sided paired $t$ test) (Fig 3C). The detected skewed hydroxymethylation at CGs is an important indicator demonstrating that hmTOP-seq can sense subtle differences in 5hmCG levels. Furthermore, the association of 5hmCG enrichment at the sense strand with gene expression levels is in good agreement with the proposed association of 5hmC with active gene expression [4]. Consistent with a suggested repressive role of 5hmC at promoter regions [4], promoters of the highly expressed genes were less enriched in 5hmCGs. Of note, the detected strand-specific 5hmCG bias persisted in the 50-ng input hmTOP-seq library (S6 Fig), suggesting that hmTOP-seq is able to discern high-resolution 5hmCG patterns in limited samples. We then compared hydroxymethylation and general cytosine modification levels at CGs across the same gene groups, using hmTOP-seq and uTOP-seq data, respectively

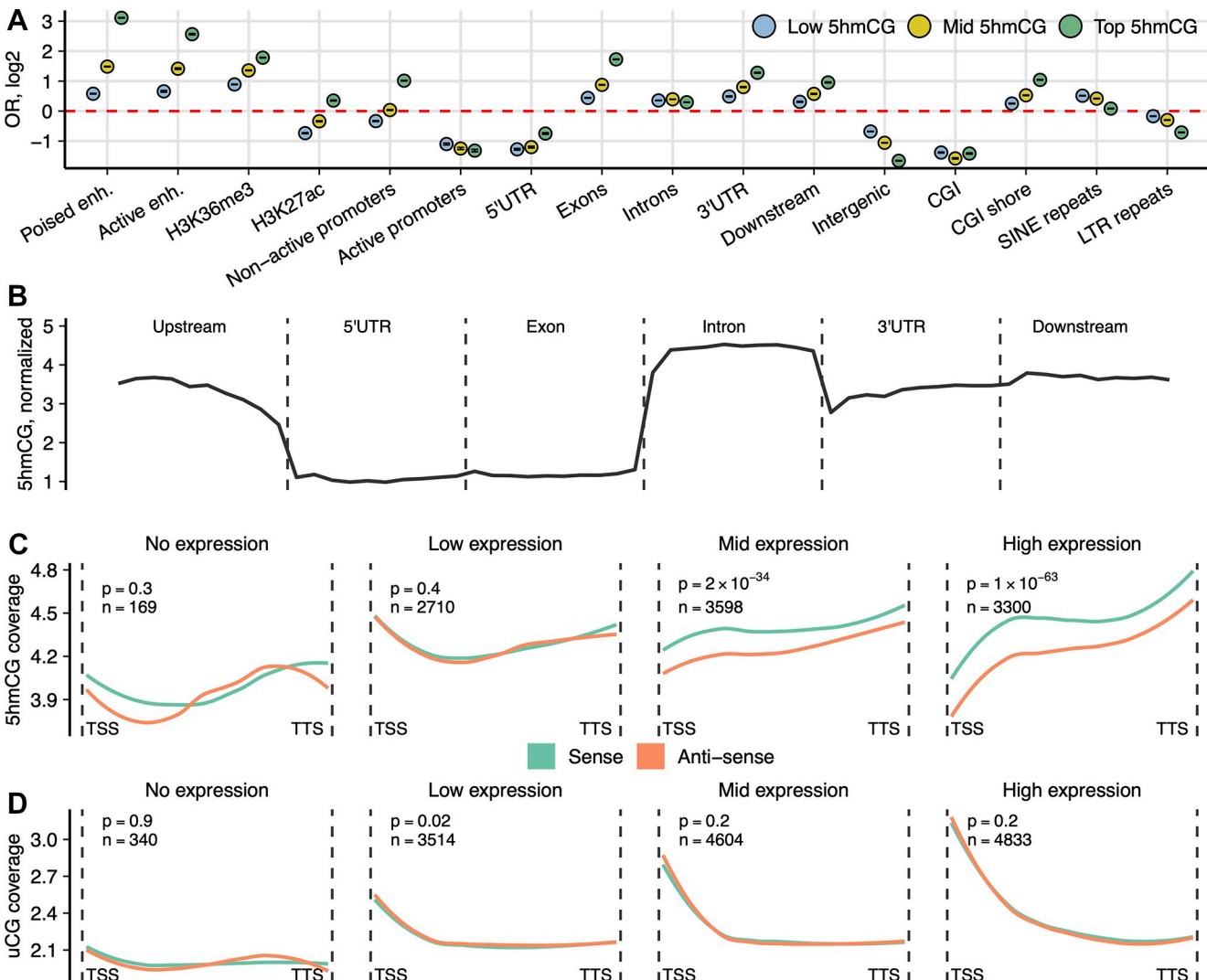

**Fig 3. Genomic distribution of 5hmCGs in mESCs.** (A) OR from Fisher's exact test for enrichment of the high- (top 20%), medium- (middle 20%), and low-coverage (bottom 20%) 5hmCGs across various genomic features. Poised enhancers ("enh."): regions with H3K4me1 mark only; active enhancers: regions with H3K4me1 and H3K27ac histone marks; active promoters: 2-kb regions upstream of the gene start that overlap H3K9ac histone mark; nonactive promoters: 2-kb upstream regions depleted in H3K9ac. All analyses are shown for a 500-ng input hmTOP-seq library. All shown enrichment values have $p \leq 1.1 \times 10^{-6}$. The data underlying this figure are included in S1 Data. (B) hmTOP-seq coverage profile normalized to CG density over different gene-associated regions: upstream (2 kb), 5′ UTR, exons, introns, 3′ UTRs, and downstream (2 kb). Distribution of (C) 5hmCGs and (D) uCGs across the sense and the antisense strands of genes grouped according to their expression level. Numbers of genes in each group and *p*-values for the modification difference between the strands are shown above each graph. All analyses are shown for a 500-ng input hmTOP-seq library. 5hmCG, hydroxymethylated CG site; CGI, CG island; H3K4me1, histone H3 lysine 4 monomethylation; H3K9ac, histone H3 lysine 9 acetylation; H3K27ac, histone H3 lysine 27 acetylation; H3K36me3, histone H3 lysine 36 trimethylation; hmTOP-seq, 5hmC-specific tethered oligonucleotide–primed sequencing; LTR, long terminal repeat; mESC, mouse embryonic stem cell; OR, odds ratio; SINE, short interspersed nuclear element; TSS, transcription start site; TTS, transcription termination site; uCG, unmethylated CG; UTR, untranslated region.

(Fig 3D). Accordingly, promoters of more highly expressed genes were relatively less methylated as compared with those of weakly expressed genes. There was no noticeable difference for the uCG distribution between the sense and the antisense strands, consistent with the symmetrical methylation of CGs [18].

To further demonstrate the capacity of hmTOP-seq for high-resolution analysis, we analyzed distribution of 5hmCGs around exon-intron boundaries. DNA hydroxymethylation and

modification differences at the exon-intron junction were reported for the immediate boundary (within the first 5 nt) and up to the first 20 nt to the boundary [17,36]. We selected all internal protein-coding exons that contained 5hmCG both on the intronic and exonic side within 25-nt distance from the splicing site. In our analysis, a cross-boundary change in 5hmCG distribution at the site of transition from exon to intron was most evident in the first 5–10 nt from the boundary on the sense strand, whereas hydroxymethylation levels in the antisense strand remained constant (Fig 4A). We identified a 5hmC peak at the −1 and −2 positions on the exonic side, then a sharp drop at around +5 and again an increase in 5hmC levels for longer distances (up to +25 nt at the intron side). Strikingly, at the site of transition from intron to exon, we detected a prominent intronic 5hmC peak at around −5 position on the coding strand. For the opposite strand, CGs that localized immediately adjacent to the intron-exon junction, as well as the peri-boundary CGs, showed a lower 5hmC level across introns relative to exons. The observed cross-boundary 5hmCG changes are in good agreement with TAB-seq data of mouse and human tissues [35], indicating that our method can approach a precision achievable only to the gold-standard methods. Importantly, the cross-boundary differences were also evident in the 50-ng input DNA hmTOP-seq analysis, in which considerably lower numbers of 5hmCGs were identified (S7 Fig). Of note, the general cross-boundary DNA modification levels of CGs followed a similar trend, as evidenced by the TOP-seq profiles of uCGs (Fig 4B).

## Discussion

To date, the TOP-seq approach, which relies on targeted nonhomologous priming of a DNA polymerase to provide enhanced mapping resolution and sequencing economy, has been implemented for analysis of unmodified [25] or hydroxymethylated CG sites in DNA ([27] and this work). Taking advantage of available empirical data, we—for the first time, to our knowledge—were able to propose a general mechanism for this unconventional mode of polymerase action. At the heart of the mechanism is proximity-driven capture of a covalently tagged target base, located at an internal position of the DNA template strand, in the active site of a DNA polymerase; the latter is attracted to the DNA by the short duplex composed of the tethered ODN and its complementary priming strand. This apparent invasion of the DNA helix requires no "jumping" of the polymerase, as has been suggested elsewhere [27]. Most importantly, the new level of mechanistic understanding enables a better evaluation of suitable tethering chemistries by simple modeling prior to empirical examination, which is exemplified by the superior resolution of the newly developed hmTOP-seq procedure as compared with its predecessor [27]. The attained precision capacitated the readout and mapping of genomic 5hmC in both CG and non-CG contexts.

The nondestructive nature of hmTOP-seq allowed the construction of 5hmC maps with less starting DNA amounts than for routine WGBS. Besides its high robustness and inherent single-base resolution, the hmTOP-seq approach offers cost-efficiency, stemming from the target-selective sequencing and facile data processing. Such analysis results in extremely informative datasets as demonstrated by whole-genome high-coverage mapping of 5hmC in mESCs using only on average 20 million of processed reads. To our knowledge, no other method to date can offer a similar performance.

We demonstrated that the method can assess subtle hydroxymethylation changes at individual CGs distributed around exon-intron cross-boundaries, the precision achievable only to the gold-standard WGBS. Furthermore, even lower-input (50 ng) DNA hmTOP-seq libraries permitted detection of strand-specific CG hydroxymethylation.

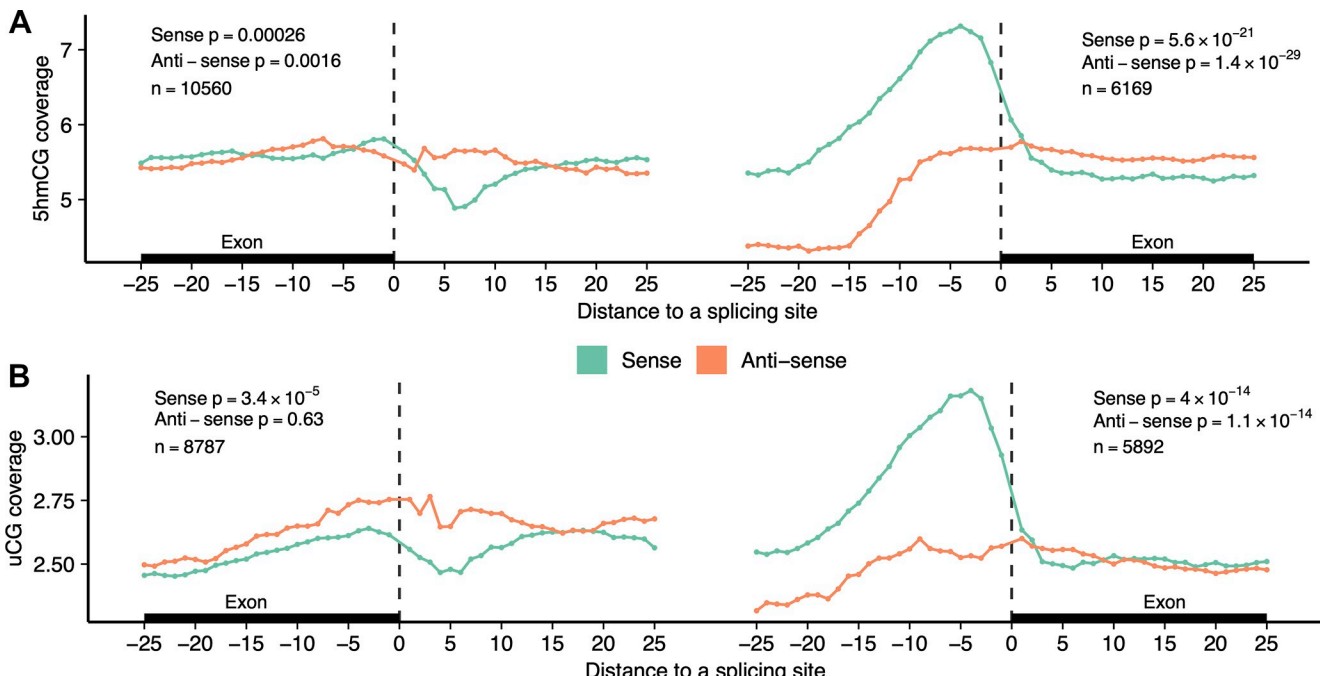

**Fig 4. CG modification profiles at exon-intron cross-boundaries.** Distribution of 5hmCGs (A) (500-ng input DNA hmTOP-seq libraries) or uCGs (B) at both sides of the exon-intron boundary is presented for the sense and the antisense strands. The x-axis shows the distance (nt) of CGs from the boundary. *p*-Values indicate a difference in coverage between exonic and intronic side of the boundary for the first 25 nt. At the exon-intron boundary (left part), general 5hmCG modification levels are higher at the exonic side as compared with the intronic side for both strands. At the intron-exon boundary (right part), the antisense strand shows the same trend, whereas the sense strand shows higher general 5hmCG levels on the intronic side. The data underlying this figure are included in S1 Data. 5hmCG, hydroxymethylated CG site; nt, nucleotide; uCG, unmodified CG site.

Most recent studies have proposed bisulfite-free methods for single-base resolution 5hmC analysis which exploit differential sensitivity of the cytosine modifications to enzymatic deamination by an AID/APOBEC family DNA deaminase [37] or chemical modification of 5hmC [38,39] to afford differential readout of 5hmC in conventional sequencing. These methods require whole-genome sequencing, which bears a hefty price tag for large-scale studies of epigenetic diseases and cancer.

hmTOP-seq, as all other methods based on the TOP-seq strategy, cannot directly determine the absolute modification levels of CGs but can infer relative hydroxymethylation of each CG based on their sequencing coverage. In addition to 5hmCG-based single-nucleotide-resolution analysis, calculation of regional h-density profiles, especially with lower-input hmTOP-seq libraries (5-ng input DNA), should be used for comparison of 5hmC levels among multiple samples. As a read-count-based epigenome profiling approach, hmTOP-seq is affected by confounders such as aneuploidy and copy number variations (CNVs). However, although for studies of heavily genetically transformed genomes, such as cancer genomes, a prior knowledge of regions affected by CNV is necessary, the presence of such large megabase regions do not prevent computation of differentially modified regions within CNVs.

Overall, hmTOP-seq is an attractive semiquantitative method for screening hydroxymethylated cytosines genome-wide in various tissues and clinical samples. We envision that the combination of inherent high resolution and cost-efficiency will pave the way for its wide application in harnessing complex human epigenome variations.

## Materials and methods

### Cell culture

Murine embryonic stem cells E14TG2a were kindly provided by Prof. Guoliang Xu (Shanghai Institute of Biochemistry and Cell Biology). mESCs were grown on 0.1% gelatin coated dishes in the presence of 1,000 U/ml LIF (ESGRO Millipore) and 15% fetal bovine serum as described previously [40].

### gDNA

DNA from mESCs was purified using standard phenol-chloroform extraction.

### Validation of hmTOP-seq in a model DNA fragment system

Two model DNA fragments were made by PCR from human *BMX* gene: 1H, 155 bp (oligonucleotides for PCR, 1H-dir 5′-TGTGTTACTGTGTGGAAAAGACC-3′, 1H-rev 5′-CCACTC CTTATAGTTTGGCTGA-3′) and 2H, 202 bp (2H-dir 5′-GCAATGTGTTGTGGAGGAGA-3′, 2H-rev 5′- CCTACTTGGGTTTGCCCTCT-3′). The 5hmC was introduced at GCGC sites of the DNA fragments by incubating 1 µg of 1H/2H DNA fragment mix with a 5-fold molar excess of M.HhaI and 13 mM formaldehyde for 1 h at room temperature followed by Proteinase K treatment (0.2 mg/ml) for 30 min at 55˚C and column purification (GeneJet PCR Purification kit [TS]) [26,30]. The efficiency of hydroxymethylation was approximately 90%, according to R.Hin6I endonuclease protection and qPCR analysis (S2E Fig).

To label 5hmC with an azide group, M.HhaI-treated 1H/2H were labeled with 10 U T4 BGT (TS) supplemented with 50 µM UDP-glucose-azide (Jena Bioscience) for 2 h at 37˚C, followed by enzyme inactivation at 65˚C for 20 min and column purification (DNA Clean & Concentrator-5, Zymo Research).

To label 5hmC-DNA with cysteamine, M.HhaI-treated 1H/2H DNA fragments were incubated with a 10-fold molar excess of M.SssI (TS), 12.5 mM cysteamine (Sigma) in 40 µl of 50 mM NaOAc (pH 6.0), 0.2 mg/ml BSA for 1 h at 30˚C, followed by enzyme inactivation at 65˚C for 15 min and column purification (GeneJet PCR Purification kit [TS]). DNA was eluted in 40 µl of 20 mM NaHCO$_3$ (pH 9.0) and supplemented with 40 µl of 0.3 M NaHCO$_3$, 20 µl Azide Amine-Activator (Interchim) to final 2 mg/ml concentration, incubated for 2 h at room temperature, and purified through GeneJet PCR Purification kit (Protocol A [TS]) columns.

Following the covalent labeling procedures, DNA was further processed as follows: after ligation of the partially complementary adapters as described previously (step 2 [25]), DNA was supplemented with 20 µM alkyne DNA oligonucleotide (ODN; 5′-T[alkyneT]TTATA-TATTTATTGGAGACTGACTACCAGATGTAACA-3′, Base-click) and 8 mM CuBr: 24 mM THPTA mixture (Sigma) in 50% of DMSO, incubated for 20 min at 45˚C, and subsequently diluted to <1% DMSO before a column purification (GeneJet NGS Cleanup kit, Protocol A [TS]).

A DBCO moiety containing ODN was produced by incubating amine DNA oligonucleotide (5′-C[T-C2-NH$_2$]TTTATATATTTATTGGAGACTGACTACTACCAGATGTAACA-3′, Metabion) with 100× excess of DBCO-sulfo-NHS ester (Glen Research) in 150 mM NaHCO$_3$ for 2 h at room temperature and subsequently purified with Oligo Clean & Concentrator (Zymo Research) spin columns. Azide-DNA was supplemented with 10 µM DBCO ODN in 25% DMSO, incubated for 2 h at 45˚C, and subsequently diluted to <1% DMSO before a column purification (GeneJet NGS Cleanup kit, Protocol A [TS]).

Coupling efficiency (S1 Table) was calculated from 2–3 independent experiments as a percentage of model DNA fragments conjugated to ODN on Agilent Bioanalyzer profiles.

A 20 μl reaction containing 5-ng ODN-conjugated DNA, 0.5 μM of a complementary priming strand (EP; 5′-TGTTACATCTGGTAGTCAGTCTCCAATAAATATAT-3′, with custom LNA modifications [Exiqon] and phosphorothioate linkages at the 3′ end), and 2.5 U Pfu DNA polymerase (TS) in Pfu buffer supplemented with 0.2 mM dNTP was incubated at the following cycling conditions: 95˚C for 2 min; 5 cycles at 95˚C for 1 min, 65˚C for 10 min, 72˚C for 10 min. Primed DNA was amplified with 2× Platinum SuperFi PCR Master Mix (TS), 0.5 μM barcoded fusion PCR primers A(Ad)-EP-barcode-primer (63 nt), and trP1(Ad)-A2-primer (45 nt) (both primers contained phosphorothioate modifications). Thermocycler conditions were 94˚C for 4 min; 15–25 cycles at 95˚C for 1 min, 60˚C for 1 min, 72˚C for 1 min. The final amplified DNA fragments were column purified (GeneJet NGS Cleanup kit, Protocol A [TS]).

To evaluate the priming efficiency, 10 μl of the priming reaction mixture containing 2.5 ng of ODN-conjugated 1H fragment was added to 50 μl mixture of Platinum SuperFi PCR Master mix, EP and 1H-dir or EP and 1H-rev primers (0.5 μM each), and 0.08x SYBR Green I (Sigma-Aldrich) and amplified on Rotor-Gene Q (Qiagen) real-time PCR machine using 30 cycles of the same thermocycling program as stated above.

## Validation of hmTOP-seq in a model DNA system

GCGC-hydroxymethylated bacteriophage lambda gDNA was prepared by treating 3 μg of fragmented lambda DNA with a 3-fold molar excess of M.HhaI and 13 mM formaldehyde for 1 h at room temperature. After reaction, DNA was purified using GeneJet PCR Purification kit (TS). Five different DNA mixtures containing 2.5%, 5%, 10%, 20%, and 40% 5hmC at GCGC sites were produced by premixing the HhaI-modified and untreated lambda DNA. Libraries were prepared as described above (with mESC gDNA) with the following changes: 300-ng DNA samples were labeled with T4 BGT as above. ODN-conjugated DNA was used in 25 μl of priming reaction mixture with 2.5 U Pfu DNA polymerase. PCR amplification (for 12 cycles) was carried out by adding all of the above reaction mixture to 100 μl of amplification reaction.

## Analysis of bacteriophage lambda hmTOP-seq data

Processing of the bacteriophage lambda hmTOP-seq sequencing data was performed as described previously [25] except for the minimal length of retained reads set at 120 nt. Processed reads were mapped to a lambda phage genome, and only the reads with mapping quality equal or above 60 and starting exactly at a CG site in a GCGC context were used for further analysis. PCR duplicates were removed using the following algorithm: for each uniquely mapped read, its start coordinate (5′ end) and read length without the 3′ adapter were obtained; reads sharing identical start coordinate and read length were considered duplicates and only one was retained.

To test the relationship between the level of hydroxymethylation of GCGC sites (2.5%, 5%, 10%, 20%, 40% 5hmC) and their coverage, a quadratic regression model was applied in which median coverage was the dependent variable and 5hmC level was the independent variable.

## Preparation of hmTOP-seq libraries of mESCs

Extracted gDNA of mESCs was sonicated on M220 Focused-ultrasonicator (Covaris) in 10 mM Tris-HCl (pH 8.5) buffer to yield fragments with a peak size of approximately 200 bp. The protocol of hmTOP-seq (Fig 1D) is as follows.

**Step 1.** The 5hmC glycosylation was carried out in a 50 μl reaction mixture with 5, 50, or 500ng of fragmented gDNA supplemented with 50 μM UDP-6-azide-glucose (Jena Bioscience) and 5 U T4 BGT (TS) for 2 h at 37˚C, followed by enzyme inactivation at 65˚C for 20 min and column purification (GeneJet PCR Purification kit [TS]).

**Step 2.** Azide-tagged DNA was end-filled using a DNA End Repair Kit (TS) according to the vendor's recommendations, and DNA was purified using the GeneJet Purification Kit (TS). A 3′-dA mononucleotide extension was added to end-repaired DNA by incubating with Klenow exo- polymerase in Klenow Buffer (TS) in the presence of 0.5 mM dATP at 37˚C for 45 min, enzyme inactivated at 75˚C for 15 min followed by purification through DNA Clean & Concentrator-5 columns (Zymo Research). Partially complementary adaptors A1/A2 (4.5 mM) (produced by annealing of partially complementary 32/33-nt oligonucleotides A1/A2, A1 5′ P-GATTGGAAGAGTGGTTCAGCAGGAATGCTGAG, and A2 5′-ACA CTCTTTCCCTACATGACAC TCTTCCAATCT) were ligated by incubating the DNA with 15 U of T4 DNA Ligase (TS) in Ligase buffer at 22˚C overnight in a total volume of 30 μl, followed by thermal inactivation at 65˚C for 10 min and column purification (DNA Clean & Concentrator-5, Zymo Research).

**Step 3.** DNA from Step 2 DNA was supplemented with 20 μM biotinylated alkyne-containing DNA oligonucleotide (5′-T[alkyneT]TTTTGTGTGGTTTGGAGACTGACTACCAGATG-TAACA-biotin, Base-click) and 8 mM CuBr: 24 mM THPTA mixture (Sigma) in 50% of DMSO, incubated for 20 min at 45˚C, and subsequently diluted to <1.5% DMSO before a column purification (GeneJet NGS Cleanup kit, Protocol A [TS]).

**Step 4.** DNA recovered after the biotinylation step was incubated with 0.1 mg Dynabeads MyOne C1 Streptavidin (TS) in buffer A (10 mM Tris-HCl [pH 8.5], 1 M NaCl) at room temperature for 3 h on a roller. DNA-bound beads were washed 2× with buffer B (10 mM Tris-HCl [pH 8.5], 3 M NaCl, 0.05% Tween 20); 2× with buffer A (supplemented with 0.05% Tween 20); and 1× with 100 mM NaCl and were finally resuspended in water and incubated for 5 min at 95˚C to recover enriched DNA fraction.

**Step 5.** Enriched DNA was subsequently used in 30 μl of priming reaction in Pfu buffer with 1.5 U Pfu DNA polymerase (TS), 0.2 mM dNTP, 0.5 μM complementary priming strand (EP; 5′-TGTTACATCTGGTAGTCAGTCTCCAAACCACACAA-3′, with custom LNA modifications [Exiqon] and phosphorothioate linkages at the 3′ end). Reaction mixture was incubated at the following cycling conditions: 95˚C for 2 min; 5 cycles at 95˚C for 1 min, 65˚C for 10 min, 72˚C for 10 min.

**Step 6.** Amplification of a primed DNA library was carried out by adding 22 μl of the priming reaction mixture to 100 μl of amplification reaction containing 50 μl of 2× Platinum SuperFi PCR Master Mix (TS) and barcoded fusion PCR primers A(Ad)-EP-barcode-primer (63 nt) and trP1(Ad)-A2-primer (45 nt) at 0.5 μM each (both primers contained phosphorothioate modifications). Thermocycler conditions were as follows: 94˚C for 4 min; 12 or 15 cycles at 95˚C for 1 min, 60˚C for 1 min, 72˚C for 1 min. The final libraries were size-selected for approximately 300-bp fragments (MagJet NGS Cleanup and Size-selection kit [TS]), and their quality was tested on Agilent 2100 Bioanalyzer (Agilent Technologies) and by qPCR (TS, Qiagen). Libraries were subjected to Ion Proton (TS) sequencing.

## Preparation of uTOP-seq libraries of mESCs

uTOP-seq libraries of mESC using 300-ng input DNA were prepared as described previously [25].

## Analysis of mESC hmTOP-seq data

hmTOP-seq data were processed as described for bacteriophage lambda hmTOP-seq with the following exceptions: a minimal raw read length was 80 nt and reads were mapped to the

mouse genome build mm10 with a minimal mapping quality of 30. The 5hmCH coverage was calculated using reads that start exactly at those CH sites that do not contain a CG closer than 7 nt in the downstream direction. h-density was calculated and TOP-seq data were processed as described previously [25]. Additionally, the h-density signal was log2 transformed.

To test correlations between the technical replicates in silico subsampling was performed by randomly selecting a fraction of reads assigned to a CG site. Correlation between hmTOP-seq and nano-hmC-Seal signal was determined in high-confidence nano-hmC-Seal regions. For each nano-hmC-Seal region, a total signal from each method was calculated and square-root transformed. Correlation between hmTOP-seq and nano-hmC-Seal as well as for hmTOP-seq technical replicates was calculated for each autosome separately.

Enrichment of genomic elements for the 5hmCG signal was calculated as follows. The 5hmCGs were divided into three signal-strength groups (low: bottom 20%; middle: mid 40%–50%; top: top 80%). Then, a contingency table was created for each CG site falling into a signal group and overlapping a genomic region. Fisher's exact test was performed to estimate the OR and $p$-value. For 5hmCH signal enrichment, the same algorithm was used, but only with one signal-strength group (all available 5hmCHs). A general gene profile was created by dividing each gene element into 10 equally sized bins and for each bin calculating average 5hmCG signal normalized by CG density. To calculate strand differences in 5hmCG and uCG along the protein-coding gene, we used the following algorithm. First, we removed genes that were shorter than 1% or longer than 99% of all the genes. Next, each gene was assigned an expression group and divided into 60 equally sized bins. Four expression groups were defined: genes without expression and genes with low, mid, and high expression divided into three equally sized groups. To have a representative signal in the gene body, we selected only those genes that had modification signal in at least 20 bins on both strands. Two-sided paired $t$ test was used to calculate differences between the strands per each expression group. To calculate profiles around the exon-intron boundaries, we selected all internal protein-coding exons that contained modification sites on both sides of the boundary (within absolute distance 1–25 nt). To test differences between the intron and exon sides, the average signal for each exon-intron boundary was calculated and a two-sided paired $t$ test was performed.

## Annotations

Mouse genome sequence (build mm10), CGI, and repeat coordinates were downloaded from the UCSC genome browser. CGI shores were defined as 2-kb regions around a CGI. Gene dataset was downloaded from the GENCODE encyclopedia of genes [41]. Gene upstream (promoter) or downstream regions were set as 2-kb regions from a given transcription start or end site, respectively. Intergenic regions were defined as regions that are 50 kb away from the protein-coding genes. mESC histone marks and RNA dataset was taken from the ENCODE data portal (ENCODE project consortium [42,43]). TAB-seq and nano-hmC-Seal datasets were obtained from the Gene Expression Omnibus (TAB-seq: GSE36173; nano-hmC-Seal: GSE77967) [18,33]. Both 5hmC modification datasets were converted to mm10 mouse genome version.

## Supporting information

**S1 Fig. Putative conformations of the tethering linkers 1a, 2a, and 3 in relation to $T_2$ (tethered ODN) and target 5hmC (gDNA) nucleotides.** Linker conformations were determined by ChemBio3D Ultra MM2 energy minimization using the target values derived from the coordinates of $dT_4$ and $dT_6$ nucleotides in the template strand of the KOD polymerase–DNA complex (PDB: 5omf) as follows: C5–C5 distance = 8.99 Å; C5-methyl bond angles for $dT_6$

(corresponds to $T_2$ in the tethered ODN) and $dT_4$ (corresponds to 5hmC) = 58.8˚ and 124.4˚, respectively; dihedral angle ($dT_4$–$dT_6$ helical twist) = 59.8˚. Actual refined values are shown in fine print. 5hmC, 5-hydroxymethylcytosine; ODN, oligodeoxyribonucleotide; PDB, Protein Data Bank.
(TIF)

**S2 Fig. Specificity and efficiency of 5hmC mapping by hmTOP-seq.** (A) Schematic view shows four priming products generated from the two model DNA fragments, 1H and 2H. Theoretical sizes of the four specific hmTOP-seq and uTOP-seq products (including 135-bp adapters) are as follows: 1H-dir 176 bp, 2H-dir 194 bp, 1H-rev 247 bp, and 2H-rev 276 bp. (B) Agilent Bioanalyzer profiles of hmTOP-seq priming products obtained from 1H and 2H model DNA fragments, each containing a single 5hmC, followed by derivatization with cysteamine (black line) or the azide group (red line). For each type of derivatization, different number of PCR cycles was required to detect comparable amounts of four products (20 cycles for azide- and 25 cycles for cysteamine-derivatization). (C) Comparison of hmTOP-seq and uTOP-seq [25] in the 1H/2H model DNA system. Bioanalyzer profiles of the products obtained from 1H and 2H model DNA fragments each containing a single 5hmC or an unmodified CG processed through the hmTOP-seq (red line) or uTOP-seq procedure (black line). In both cases, the corresponding workflow generates four specific products with similar efficiencies (15 cycles of PCR were used). Azide-labeling of unmethylated CG sites in the model DNA fragments was performed as described [25]. (D) Agilent Bioanalyzer profiles of hmTOP-seq priming products obtained from 1H/2H, followed by copper-free (black line) or Cu(I)-catalyzed (red line) click conjugation to DNA oligonucleotide. In both cases, 25 cycles of PCR were used. (E) Assessment of the M.HhaI-directed hydroxymethylation efficiency on two model DNA fragments. After incubation of 1H/2H model DNA fragments with M.HhaI in the presence of formaldehyde, the DNA fragments were cleaved with Hin6I restriction endonuclease and the amount of uncleaved DNA was evaluated by qPCR with the respective primer pairs (1H-dir/1H-rev or 2H-dir/2H-rev; see "Validation of hmTOP-seq in a model DNA fragment system" in Materials and methods). The data underlying this section are included in S2 Data. 5hmC, 5-hydroxymethylcytosine; A1 and A2, strands of a partially complementary adaptor; Ad-A2 and Ad-TO, adapters containing NGS platform-specific 5′-end sequences; hmTOP-seq, 5hmC-specific TOP-seq; qPCR, quantitative PCR; TO, tethered oligodeoxyribonucleotide; uTOP-seq, uCG-specific TOP-seq.
(TIFF)

**S3 Fig.** Distance distribution of read start positions from (A) a nearest GCCG or (B) a CG site in the hmTOP-seq library of pre-hydroxymethylated lambda DNA (2.5% 5hmC at GCGC sites) and mESCs, respectively. The data underlying this figure are included in S2 Data. 5hmC, 5-hydroxymethylcytosine; hmTOP-seq, 5hmC-specific tethered oligonucleotide–primed sequencing; mESC, mouse embryonic stem cell.
(TIFF)

**S4 Fig. Correlation between technical replicates using subsamples of the hmTOP-seq data.** Sequencing reads were sampled from 500-ng and 50-ng input DNA hmTOP-seq libraries, respectively. The data underlying this section are included in S2 Data. hmTOP-seq, 5hmC-specific tethered oligonucleotide–primed sequencing.
(TIFF)

**S5 Fig. hmTOP-seq analysis of non-CG sites.** Odds ratio (Fisher's test) for enrichment (A) or distribution (B) of 76,665 5hmCHs detected in mESCs across various genomic features. All enrichments have $p < 0.05$. The data underlying this section are included in S2 Data. 5hmCH,

hydroxymethylated CH site; hmTOP-seq, 5hmC-specific tethered oligonucleotide–primed sequencing; mESC, mouse embryonic stem cell.
(TIFF)

**S6 Fig. Genomic distribution of 5hmCGs in mESCs.** Distribution of 5hmCGs in 50-ng input DNA hmTOP-seq libraries across the sense and the antisense strands of genes grouped according to their expression level. Numbers of genes in each group and *p*-values for the modification difference between the strands are shown above each graph. hmTOP-seq, 5hmC-specific tethered oligonucleotide–primed sequencing; mESC, mouse embryonic stem cell.
(TIFF)

**S7 Fig. Changes in 5hmCG distribution at exon-intron cross-boundaries.** Distribution of 5hmCGs in 50-ng input DNA hmTOP-seq libraries at both sides of the exon-intron boundary is presented for the sense and the antisense strands. The x-axis shows the distance (nt) of CGs from boundary. *p*-Values indicate a difference in coverage between exonic and intronic side of the boundary for the first 25 nt. The data underlying this section are included in S2 Data. 5hmCG, hydroxymethylated CG site; hmTOP-seq, 5hmC-specific tethered oligonucleotide–primed sequencing; nt, nucleotide.
(TIFF)

**S1 Table. Structural parameters and TOP-seq performance of tethering linkers.** TOP-seq, tethered oligonucleotide–primed sequencing.
(XLSX)

**S2 Table. Sequencing statistics of hmTOP-seq and uTOP-seq libraries of mESC DNA.** hmTOP-seq, 5hmC-specific TOP-seq; mESC, mouse embryonic stem cell; uTOP-seq, uCG-specific TOP-seq.
(XLSX)

**S1 Data. Numerical data underlying Figs 2A, 2B, 2C, 2D, 2E, 2F, 2G, 3A, 4A and 4B.**
(XLSX)

**S2 Data. Numerical data underlying S2E, S3A, S3B, S4, S5A, S5B and S7 Figs.**
(XLSX)

## Acknowledgments

We are grateful to Janina Ličytė for initial experiments of 5hmC labeling with BGT and gDNA of mESCs and Prof. Viktoras Masevičius for advice on MM2 modeling. We thank Prof. Guo-liang Xu for mESCs and Dr. Vaidotas Stankevičius for cultivating them.

## Author Contributions

**Conceptualization:** Saulius Klimašauskas, Edita Kriukienė.

**Formal analysis:** Saulius Klimašauskas.

**Funding acquisition:** Saulius Klimašauskas, Edita Kriukienė.

**Investigation:** Povilas Gibas, Milda Narmontė, Zdislav Staševskij, Saulius Klimašauskas, Edita Kriukienė.

**Methodology:** Povilas Gibas, Milda Narmontė, Zdislav Staševskij, Juozas Gordevičius.

**Project administration:** Edita Kriukienė.

**Software:** Povilas Gibas.

**Supervision:** Edita Kriukienė.

**Visualization:** Povilas Gibas, Milda Narmontė.

**Writing – original draft:** Edita Kriukienė.

**Writing – review & editing:** Povilas Gibas, Milda Narmontė, Juozas Gordevičius, Saulius Klimašauskas, Edita Kriukienė.

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
