## [Editor Report · Decision Letter 0]

8 Nov 2019

Dear Dr Kriukiene, 

Thank you for submitting your manuscript entitled "Precise Genomic Mapping of 5-hydroxymethylcytosine via Covalent Tether-directed Sequencing" for consideration as a Research Article by PLOS Biology. Thank you also for your patience as we completed our editorial process, and please accept my apologies for the delay in providing you with our decision.

Your manuscript has now been evaluated by the PLOS Biology editorial staff as well as by an academic editor with relevant expertise and I am writing to let you know that we would like to send your submission out for external peer review.

Please re-submit your manuscript within two working days, i.e. by Nov 12 2019 11:59PM.

Kind regards,

Ines

--

Ines Alvarez-Garcia, PhD

Senior Editor

PLOS Biology

Carlyle House, Carlyle Road

Cambridge, CB4 3DN

+44 1223–442810

---

## [Decision Letter · Decision Letter 1]

20 Dec 2019

Dear Dr Kriukiene,

Thank you very much for submitting your manuscript "Precise Genomic Mapping of 5-hydroxymethylcytosine via Covalent Tether-directed Sequencing" for consideration as a Methods and Resources at PLOS Biology. Thank you also for your patience as we completed our editorial process, and please accept my apologies for the delay in providing you with our decision. Your manuscript has been evaluated by the PLOS Biology editors, an Academic Editor with relevant expertise, and by three independent reviewers.

As you will see, the reviewers are positive in general and find the method described in the manuscript interesting and novel. However, they also raise several issues that need to be addressed before we can consider the manuscript for publication. After consulting with the academic editor, we would like to highlight several points, which we deem particularly important and consider essential. These include issues relating to priming efficiency (Rev. 2, point 1), yield of reaction (Rev. 2, point 2), demonstration of b-GT activity outside GC (Rev. 2, point 5) and demonstration of a clear benchmark (Rev. 3, point 1). In addition, inclusion of a discussion of limitations of the method, as suggested by Reviewer 1, will also be important.

In light of the reviews (attached below), we will not be able to accept the current version of the manuscript, but we would welcome re-submission of a revised version that takes into account the reviewers' comments. We cannot make any decision about publication until we have seen the revised manuscript and your response to the reviewers' comments. Your revised manuscript is also likely to be sent for further evaluation by the reviewers.

We expect to receive your revised manuscript within 2 months. 

**IMPORTANT - SUBMITTING YOUR REVISION**

*NOTE: In your point by point response to to the reviewers, please provide the full context of each review. Do not selectively quote paragraphs or sentences to reply to. The entire set of reviewer comments should be present in full and each specific point should be responded to individually, point by point.

*Re-submission Checklist*

*Published Peer Review*

*PLOS Data Policy*

*Blot and Gel Data Policy*

Sincerely,

Ines

--

Ines Alvarez-Garcia, PhD

Senior Editor

PLOS Biology

Carlyle House, Carlyle Road

Cambridge, CB4 3DN

+44 1223–442810

Reviewers’ comments

Rev. 1: Sriharsa Pradhan – please note that this reviewer has waived anonymity

This manuscript describes a new sequencing technique called hmTOP-seq that can be used for identifying 5hmC at single base resolution genome-wide. This method involves Fragmented genomic DNA is tagged with an azide group through BGT-glucosylation flowed by ODN containing single biotin group is tethered to azide-groups using click chemistry. The biotinlabeled fragments are captured on streptavidin beads, TO-primed strand extension and PCR amplification was performed to generate sequencing library. The authors hypothesized that this method would identify 5hmC on any genomic DNA. To validate their approach, the authors used mouse ES genomic DNA and compared their sequencing data with data obtained from TAB-seq (bisulphite based) and ABA-seq and Pvu-Seal Seq (restriction enzyme based) and found good correlation between 5hmC mapped regions. The total number of the hmTOPseq called 5hmCH sites constituted 1.6% of 5hmCGs similar to 1.3% reported by TABseq. There are some difference between restriction enzyme based methods. The authors have also compared 5hmC distributions at other epigenomics features such as histone modifications and genic distribution. Based on their analyses, the authors conclude that the non-destructive nature of hmTOP-seq allowed the construction of 5hmC maps with several hundred-fold less starting DNA amounts than for WGBS, thus is a cost-effective target enrichment protocol. Furthermore, only on average 20 M of processed reads can be used for mES genome 5hmC analysis. This is a good and through study supported by chemistry and pfu enzymatic amplification characteristics. The method described is likely to be used by various labs studying DNA hydroxymethylation and function. I do not have any major criticisms few minor clarifications needed this manuscript’s readability better.

Minor comments:

1. Would the authors mention the shortcomings of this method? In general, all protocols have some issues that needs to be performed carefully for optimal results.

2. Similarly, a stepwise working protocol i.e. easy to follow must be included in supplementary section for reproducibility of the protocol.

Rev. 2:

Gibas et al, have developed a new method to sequence and map 5hmC, an important modified DNA mark found in humans. Their technique relies on enzymatic modification of 5hmC with an azido-containing sugar, then conjugation to an oligonucleotide primer. Following, enrichment of the oligo-containing DNA, this primer is used to amplify the surrounding region of the 5hmC. Excitingly, they show the first base extension of this priming occurring mostly at the 5hmC it is attached to. This gives them the ability to map the position of 5hmC at single-base resolution, not currently possible with pull-down techniques. I believe this is an interesting piece of work and deserves to be published, however there are a number of pieces of information missing to confirm their findings before this should happen.

Major comments:

1. The initial analysis of the technique on model DNA fragments is weak and very badly explained. The authors must re-explain what is happening in Figure S2, as it is not clear. This is vital as it underpins their whole technology. It is important for them to sequence the amplified products to find out what percentage actually prime at the 5hmC site verses the neighbouring sites. Additionally, the use of ‘high’, ‘medium’, and ‘low’ in Table S1 is unacceptable. The authors must quantify their findings. For instance, in the text they mention the DBCO addition ‘failed to generate any substantial amounts of the priming’, but in Table S1 they mark it as ‘low’.

2. The analysis of the technique on the bacteriophage was also lacking. How does formaldehyde and bacteriophage genomic DNA form 5hmC? Did they analyse the DNA to confirm the yield of this reaction? They authors seem to assume 100% yield. Why was analysis only considered at GCGC sites? The authors should have again mapped the priming sites to see if this matches the analysis of the synthetic DNA. Additionally, in Fig 2B, why was a curved line used as a trend line, this looks linear through the medians?

3. Regarding the analysis of the mESCs data, the quantification of mapped reads around CG sites is vague. In the text it says 93% ‘at or around CGs’, why can’t this be quantified to % at each position relative to the 5hmC. By the Fig 2C this looks to be ~65% at the 5hmC site. This should then be compared to the data from the model DNA and bacteriophage mapping, which was not conducted. I think I understand, but the authors should really explain why their method can analyse asymmetric 5hmC so well (I believe because the direction of priming has to be 5’ to 3’). It would be good to make a figure for this.

4. A lot of the time is unclear if the mESC data is being analysed with only the 500 ng library, or combined with either or both of the 50 and 5 ng libraries. This needs to be clear at all times. How many replicates of each library size were prepared and analysed? The pearson correlation coefficient is good for the 500 ng library, but poor for the 50 ng library and no correlation is seen for the 5 ng library. This indicates that they require at least 500 ng input DNA for good coverage of 5hmC (even if less of the library can be sequenced following this). This goes against one of the main advantages the authors are claiming. Additionally, how good is it to then use data from the 50 and 500 ng libraries together?

5. The authors analyse 5hmC in non-CG context with their method. However, has it been previously shown that the b-GT is active outside of the CG context? If not, the authors should quantify this activity in synthetic DNA to confirm their method is actually functional in these contexts.

Other comments:

6. More citations are required where reference 1 is, to highlight the amount of work that has been carried out into quantifying the levels of 5hmC across multiple tissue types.

7. 5hmC is only mentioned as an intermediate in the demethylation pathway, please include the extra details that demonstrate it may be an epigenetic mark in its own right.

8. The authors mention that high sequencing depths is ‘prohibitively expensive’ (page 2). I think this is unfair, as many groups do sequence to high depths. Prohibitively is not the correct term.

9. As the authors use the ‘TOP-seq’ method also, they need to explain how this works.

10. The ‘h-density’ is not explained at all, and I do not know what this term refers to. Please explain why they have used this, and what the ‘180 bp window’ is. Does this remove the single base resolution advantage?

11. At the end of page 8 the authors say they compare ‘hydroxymethylation and general cytosine modification levels’ using hmTOP and TOP. How was this done as the authors haven’t demonstrated that TOP-seq can show where modified cytosines are present, just where non-modified cytosine is present. I don't see how they can call a modified cytosine from the negative of TOP-seq, as their method is not truly quantifiable.

12. The authors mention reference 18 in passing. However, this is an incredibly important literature reference, as they perform almost the same method described within. This should be described in more detail. From reading reference 18, it does seem this paper has a major advantage of closer to single-base resolution, but this needs to be discussed in more detail.

13. Why was LNA and phosphorothioate modifications used on the priming strand (page 14)?

Rev. 3:

In this study Gibas et al set a goal to develop a method to map 5-hydroxymethylcytosine (5hmC) in DNA. Previously, the group demonstrated that tethering of non-complimentary oligonucleotides to unmodified CpGs enables priming of DNA polymerases. As a consequence, loci containing unmodified CpGs can be enriched by amplification and identified using short read sequencing (TOP-seq).

To map genome-wide 5hmC authors explore several strategies. DNA methyltransferase SssI and β-glucosyltransferase (BGT) were used to modify 5'hydroxymethyl in 5hmC with functional groups, which can be chemically ligated to oligonucleotides. The best efficiency was achieved by employing BGT, which installed azidoglucose enabling azide-alkyne cycloaddition (CLICK). Oligonucleotide complementary primers are then used to amplify and enrich 5hmC containing DNA, which is sequenced and mapped to localise the modification (hmTOP-seq). To evaluate the performance of the method authors sequence in vitro hydroxymethylated phage DNA and mouse embryonic stem cell DNA. Sequencing results demonstrated near-single nucleotide resolution - 60% reads exactly start with a CpG, whereas 93% of reads are localised within 4 bp. Important advantage of the hmTOP-seq is that the method works with up to 5 ng of DNA. However, the performance of the method, when starting with 500 ng of DNA is substantially better achieving twice better correlation between the replicated and the coverage of 5hmCpGs. Finally, authors were able to detect both 5hmCpG and 5hmCpH sequences, strand-specific enrichment of 5hmCpGs in mES cell genome and enrichment of 5hmCs around the splice junction, demonstrating sensitivity of the method.

A number of 5hmC mapping techniques already have been published and are in use. Similar approach to install oligonucleotides for 5hmC mapping was used by Hu et al (2019, Jump-seq). Authors of this manuscript, however did employ different oligonucleotides and managed achieve a better resolution. Overall, I think the manuscript does provide convincing proofs that priming based 5hmC mapping strategy could be successfully used to map 5hmCs at a lower cost to other methods.

I have following criticisms, which need to be addressed before publication:

1. It is important to benchmark performance of hmTOP-seq in ability to detect 5hmC sites with different modification frequency. Authors could use TAB-seq data to select sites with different degrees of 5hmC and examine correlation with hmTOP-seq coverage.

2. What are the sources of the inefficiency of hmTOP-seq - is it labelling of 5hmCs or DNA polymerase priming from non-complimentary oligonucleotide?

3. Authors state - "Using six hmTOP-seq control libraries we observed 55,025 false-positive sites". I was not able to find what are the cut-offs for false positives and what criteria were used to define them.

4. When discussing strand bias instead of term "opposite" strand, "template" strand should be used.

---

## [Decision Letter · Decision Letter 2]

12 Feb 2020

Dear Dr Kriukiene,

Thank you for submitting your revised Methods and Resources manuscript entitled "Precise Genomic Mapping of 5-hydroxymethylcytosine via Covalent Tether-directed Sequencing" for publication in PLOS Biology. I have now obtained advice from two of the original reviewers and have discussed their comments with the Academic Editor. 

Based on the reviews (attached below), we will probably accept this manuscript for publication, assuming that you will modify the manuscript to address the two remaining points raised by Reviewer 2. Please also make sure to address the data and other policy-related requests noted at the end of this email.

We expect to receive your revised manuscript within two weeks. Your revisions should address the specific points made by each reviewer. In addition to the remaining revisions and before we will be able to formally accept your manuscript and consider it "in press", we also need to ensure that your article conforms to our guidelines. A member of our team will be in touch shortly with a set of requests. As we can't proceed until these requirements are met, your swift response will help prevent delays to publication.

*Copyediting*

*Published Peer Review History*

*Early Version*

*Submitting Your Revision*

Sincerely,

Ines

--

Ines Alvarez-Garcia, PhD

Senior Editor

PLOS Biology

Carlyle House, Carlyle Road

Cambridge, CB4 3DN

+44 1223–442810

DATA POLICY:

Fig. 2A, B, C, D, E, F, G; Fig. 3A; Fig. 4A, B; Fig. S2E; Fig. S3A, B; Fig. S4; Fig. S5A, B and Fig. S7

Reviewers’ comments

Rev. 2:

The authors have substantially improved their manuscript. I only have two further comments:

1) Regarding the trend line in Fig 2B. After plotting a linear regression, this also does not look as though it fits. Potentially it is linear from 0-20, but then it plateaus. I didn't mean for the authors to just put a straight line on the graph. They should actually use some statistics to identify the trend.

2) The authors mention input quantity as an improvement over bisulfite sequencing in the introducion "demands sizeable amounts of input DNA" and the discussion "hmTOP-seq allowed the construction of 5hmC maps with several hundred-fold less starting DNA amounts than for WGBS". The authors method does have some distinct advantages over WGBS commonly, however input quantity is not one of them and 'hundred-fold less' is grossly overstating even to the conventional bilsufite sequencing in input quantities (0.5-5 μg DNA) (DOI 10.1186/s13059-018-1408-2). At this reference, you can also find links to papers where WGBS has been performed on a few hundred cells to even single cells. In the authors manuscript 5 ng could not be used for single base resolution.

Rev. 3:

The authors addressed my concerns. The manuscript is now suitable for publication.

---

## [Editor Report · Decision Letter 3]

27 Mar 2020

Dear Dr Kriukiene,

On behalf of my colleagues and the Academic Editor, Tom Misteli, I am pleased to inform you that we will be delighted to publish your Methods and Resources in PLOS Biology. 

Early Version

PRESS 

Kind regards,

Alice Musson

Publication Assistant, 

PLOS Biology

on behalf of

Ines Alvarez-Garcia,

Senior Editor

PLOS Biology